# The Influence of the Preparation Method on the Physico-Chemical Properties and Catalytic Activities of Ce-Modified LDH Structures Used as Catalysts in Condensation Reactions

**DOI:** 10.3390/molecules26206191

**Published:** 2021-10-14

**Authors:** Alexandra-Elisabeta Stamate, Rodica Zăvoianu, Octavian Dumitru Pavel, Ruxandra Birjega, Andreea Matei, Marius Dumitru, Ioana Brezeștean, Mariana Osiac, Ioan-Cezar Marcu

**Affiliations:** 1Department of Organic Chemistry, Biochemistry and Catalysis, Faculty of Chemistry, University of Bucharest, 030018 Bucharest, Romania; alexandra-elisabeta.stamate@drd.unibuc.ro (A.-E.S.); octavian.pavel@chimie.unibuc.ro (O.D.P.); 2Research Center for Catalysts and Catalytic Processes, Faculty of Chemistry, University of Bucharest, 030018 Bucharest, Romania; 3National Institute for Lasers, Plasma and Radiation Physics, 077125 Magurele, Romania; ruxandra.birjega@inflpr.ro (R.B.); andreea.purice@inflpr.ro (A.M.); marius.dumitru@inflpr.ro (M.D.); 4National Institute for R&D of Isotopic and Molecular Technologies, 400293 Cluj-Napoca, Romania; ioana.brezestean@itim-cj.ro; 5Biomolecular Physics Department, Faculty of Physics, Babes-Bolyai University, 400084 Cluj-Napoca, Romania; 6Department of Physics, INCESA, University of Craiova, 200585 Craiova, Romania; mariana71osiac@gmail.com

**Keywords:** layered double hydroxides, cerium, mechanochemical, co-precipitation, cyclohexanone self-condensation, Claisen–Schmidt condensation, chalcone, flavone

## Abstract

Mechanical activation and mechanochemical reactions are the subjects of mechanochemistry, a special branch of chemistry studied intensively since the 19th century. Herein, we comparably describe two synthesis methods used to obtain the following layered double hydroxide doped with cerium, Mg_3_Al_0.75_Ce_0.25_(OH)_8_(CO_3_)_0.5_·2H_2_O: the mechanochemical route and the co-precipitation method, respectively. The influence of the preparation method on the physico-chemical properties as determined by multiple techniques such as XRD, SEM, EDS, XPS, DRIFT, RAMAN, DR-UV-VIS, basicity, acidity, real/bulk densities, and BET measurements was also analyzed. The obtained samples, abbreviated HTCe-PP (prepared by co-precipitation) and HTCe-MC (prepared by mechanochemical method), and their corresponding mixed oxides, Ce-PP (resulting from HTCe-PP) and Ce-MC (resulting from HTCe-MC), were used as base catalysts in the self-condensation reaction of cyclohexanone and two Claisen–Schmidt condensations, which involve the reaction between an aromatic aldehyde and a ketone, at different molar ratios to synthesize compounds with significant biologic activity from the flavonoid family, namely chalcone (1,3-diphenyl-2-propen-1-one) and flavone (2-phenyl-4H-1benzoxiran-4-one). The mechanochemical route was shown to have indisputable advantages over the co-precipitation method for both the catalytic activity of the solids and the costs.

## 1. Introduction

Layered double hydroxides (LDHs), a class of anionic compounds which can be found in nature as minerals and can also be synthesized, are currently generating increasing interest among scientists. Due to their particular lamellar structure and composition flexibility, these synthetic inorganic materials exhibit unique properties that make them suitable for various applications. Nevertheless, their main application area is catalysis, either as precursors, catalyst supports, or actual catalysts [1].

Structurally, layered double hydroxides can be described as lamellar compounds similar to brucite, Mg(OH)_2_. In the latter, every Mg^2+^ ion is octahedrally surrounded by six hydroxyl ions and shares edges to form infinite sheets. These sheets overlap, yielding a layered network held together by hydrogen bonds or van der Waals forces [2]. If a part of the Mg^2+^ ions is replaced by trivalent cations having a similar ionic radius, such as Al^3+^ or Fe^3+^, the entire layer will have a positive charge density. The electrical neutrality of the resulting structure is maintained by the anions which occupy the interlayer space. The following general molecular formulas describe layered double hydroxides: [M^(II)^_1–x_M^(III)^_x_(OH)_2_][A^m–^_x/m_·nH_2_O] or [M^(I)^_1–x_M^(III)^_x_(OH)_2_][A^m–^_(2x+1)/m_·nH_2_O] [3], where M^I^ (Li), M^II^ (Mg, Ni, Co, Zn), M^III^ (Al, Cr, Fe, Co) are metallic cations connected through the hydroxyl groups, in a cation-containing layer; A^m–^ represents the anions which, along with *n* molecules of water, form the anionic layer; and *x* is the charge density or the anionic exchange capacity, which takes values between 0.15 and 0.5 [4,5]. For their synthesis and post-synthesis modifications, various well-established solution-based methods have been developed. The most commonly used are simple co-precipitation methods such as precipitation at low and high supersaturation and urea hydrolysis [6]. However, these conventional methods have the following serious drawbacks: the processes are long-lasting, accurate pH control and subsequent heating are needed, and there is an increased consumption of energy during the ageing step, which requires significant time [6].

Recently, a new synthesis method based on mechanochemical treatment yielding related intercalates has been developed [7,8,9]. Apart from the evident advantage of easy operation, this approach shows great potential to overcome the difficulties associated with solution operation, such as the different precipitation rates with pH adjustment [10]. To our knowledge, several studies have already been devoted to the synthesis of LDHs modified with different rare-earth elements through mechanochemical techniques [11,12]. Still, none of them have reported a LDH catalyst modified with cerium.

LDHs present a promising future in the catalysis domain due to their unique properties, such as variable and tunable composition, high specific area, easy preparation, and many others. Moreover, the mixed oxides derived from the calcination of the layered double hydroxides are known to have increased basicity generated by strong Lewis base pairs, like O^2–^-M^n+^ (M-metal). 

An interesting chemical transformation where the base catalytic properties of the LDHs/mixed oxides can be widely explored is the condensation reaction through which many types of compounds of vital interest for different kinds of industries can be synthesized. For instance, the self-condensation reaction of cyclohexanone, which can be catalyzed by acidic or basic catalysts [13], has an essential role in caprolactam and adipic acid production processes and the production of 2-phenylphenol. In the cyclohexanone self-condensation, the adduct 1′-hydroxy-[1,1′-bicyclohexyl]-2-one is first generated, being subsequently transformed into a mixture of 2-(1-cyclohexen-1-yl)cyclohexanone and 2-cyclohexylidencyclohexanone by dehydration. The presence of another cyclohexanone molecule in the reaction medium can push the transformation to the di-condensation product or even further to dodecahydrotriphenylene or trihydroxyperhydrotriphenylene. Nowadays, although some catalytic materials have been highlighted for this transformation, including Mg-Al mixed oxide and reconstructed hydrotalcite [14], zirconia [15], MOF-encapsulating phosphotungstic acid [16], amberlyst 15 [13,17], silica chloride [18], keggin-type lanthanum phosphotungstate [19], and ion exchange resin [20], a lack of data involving the use of Ce-modified LDH catalysts can be noticed in the literature. Other than the self-condensation process, another interesting condensation reaction is the Claisen–Schmidt condensation, where compounds belonging to the flavonoid class, namely chalcone and flavone, are synthesized. Chalcone (CH) is an aromatic ketone and an enone that forms the central core for various important compounds, having various biological activities such as being anti-diabetic, anti-neoplastic, anti-hypertensive, anti-retroviral, anti-inflammatory, anti-parasitic, anti-histaminic, anti-malarial, antioxidant, anti-fungal, and so on [21,22,23]. Similarly, flavone (FL) is considered the “backbone” of many compounds used to treat numerous diseases. The literature on this topic indicates that various LDH compositions were tested as catalysts for chalcone and flavone synthesis via the Claisen–Schmidt condensation reaction [24,25,26]. Still, no studies have referred to the usage of LDH-type compounds modified with cerium until now. 

Considering the aspects mentioned above, in the current paper we intend to present the influence of the preparation method (co-precipitation and mechanochemical) on the physico-chemical properties and the catalytic activity of Ce-modified LDH structures and their corresponding mixed oxides in the above-mentioned condensation reactions.

## 2. Results and Discussion

### 2.1. XRD Analysis

The XRD patterns (Figure 1a) show that both the dried form of the precipitated (HTCe-PP) and the mechanochemical (HTCe-MC) synthesized samples containing Ce expose the typical reflections of a layered double hydroxide-hydrotalcite type-phase (HT) (Mg_6_Al_2_(CO_3_)(OH)_16_·4H_2_O, ICDD card 054–1030) as the dominant phase and small lines corresponding to CeO_2_-phase (ICDD card 034-0394) as a minor phase. They are Miller indexed on Figure 1a. Reflections corresponding to the cerium oxycarbonate side phase (Ce_2_(CO_3_)_2_O·H_2_O, ICDD card 044-0617), denoted with (*) on Figure 1a, were noticed only in the diffraction pattern of HTCe-PP.

The XRD patterns of the calcined form of the cerium-modified samples presented in Figure 1b display the reflections of both the mixed oxide Mg(Al)O periclase-type phase and the CeO_2–_cerianite phase. The structural data for the dried and calcined materials are gathered in Table 1 and Table 2, respectively. For the hydrotalcite-type phase, two crystallite dimensions were calculated: the crystallite size perpendicular to the brucite-like layers (D_003_) and the crystallite size parallel to the brucite layers (D_110_). For the oxide phases, the most intense peaks from each phase were used. From the data presented in Table 1, a slight increase of the a-parameter related to the minimum cation–cation distance in the brucite-like layer is evidenced. This is most likely due to an increase of the Mg/Al ratio, in comparison with Mg/Al = 3 for the standard ICDD card, because Ce^3+^, having a very large 6-coordinated ionic radius (1.01 Å) compared to Mg^2+^ (0.72 Å) and Al^3+^ (0.535 Å), almost certainly fails to enter in the brucite-like layer. In balance, a fluorite-type CeO_2_ phase with unit cell parameter values almost identical to the standard cerianite (ICDD card 034-0394) occurred in both the HTCe-PP and HTCe-MC samples, along with an oxycarbonate phase for the HTCe-PP. The crystallite sizes related to the brucite-layer and the CeO_2_ phases are smaller for HTCe-MC than for HTCe-PP, indicating that the mechanochemical synthesis induces a slight degree of structural disorder. The slightly smaller dimensions of the crystalline phases of the mechanochemical prepared sample compared to the sample prepared via co-precipitation are preserved in the calcinated form of the materials (Table 2). As expected, the a-parameter values of the Mg(Al)O phases are smaller than those of the standard periclase MgO due to the presence of Al^3+^ substituting cations. The proportion of cerianite phase in the calcined form of the samples is higher and their unit cell parameters are smaller, which might be due to the substituting insertion of Mg^2+^ or Al^3+^ generating defective and, probably, more active ceria phases in comparison with the as-prepared samples they originated from.

### 2.2. SEM Analysis

The SEM morphology (Figure 2a–d) of the powders reveals a hierarchical structure with different aspects for the as-prepared samples compared to their corresponding calcined forms.

For the as-synthesized samples, assemblies of nanoplates nearly perpendicular to the outer surfaces of microspheres, characteristic of LDH structures, appeared, while for the calcined samples, the plate shape turned into a cloud-like form, suggesting that the hydrotalcite-like structure of the LDH precursor was destroyed during the thermal treatment. The SEM images of HTCe-MC and Ce-MC show agglomerations of smaller particles compared to the samples obtained by precipitation method. This fact may be related to the absence of the ageing step in the mechanochemical preparation, which shortened the crystal growth duration and hence the size of the crystallites. 

### 2.3. Chemical Composition

The chemical compositions of the samples were determined via two different spectral techniques: energy-dispersive X-ray spectroscopy (EDS) and X-ray photoelectron spectroscopy (XPS), respectively. The difference between the two techniques is, in a concise description, related to their relative “depth of analysis”. For EDS, the depth is around μm, giving the “bulk” concentration of chemical elements similar to other bulk sampling techniques, such as XRF or AAS. On the other hand, the depth penetration of the signal for XPS comes from the near-surface around 2–3 nm and, hence, it gives information on the elements present on the surface of the samples, including their oxidation state, which is useful in heterogeneous catalysis, adsorption, corrosion, or adhesion studies. The high-resolution XPS Ce3d spectra (between 870 and 930 eV) of the four samples (Appendix A) could be deconvoluted into 10 bands, divided into 2 multiplets, labeled as υ and u, corresponding to the spin-orbit coupling of 3d 5/2 and 3d 3/2, respectively. The bands denoted as υ (u), υ″ (u″), and υ‴ (u‴) are ascribed to the photoemissions from the Ce^4+^ 3d core levels, respectively, while the signals υ^0^ (u^0^) and υ′ (u′) are assigned to photoemissions from Ce^3+^ cations [27,28]. The proportion of Ce^3+^ was calculated by comparing the integrated intensities of the peaks of υ, υ″, υ‴, u, u″, and u‴ of Ce^4+^ and υ^0^, υ′, u^0^, and u′ of Ce^3+^ in the XPS spectrum [29]. The chemical data are presented in Table 3. The data examination shows a decrease of the Mg/Al ratio on the external near-surface of the samples in comparison with the “bulk” values for all four catalysts. 

The Ce/Al ratio values are quite dispersed, probably due to a certain degree of the inhomogeneity of the samples. The proportion of surface Ce^3+^ ions is high for all the samples, being much higher than that of commercial ceria powders, such as a 4 N nanopowder (Ce^3+^/Ce^4+^ = 0.45) and an abrasive powder (Ce^3+^/Ce^4+^ = 0.34) [29]. The mixed Ce^3+^ and Ce^4+^ valence state of the fluorite structure of CeO_2_ is considered to be responsible for many properties of ceria in various fields [29]. 

It is noteworthy that the surface Mg/Al ratio decreases for both LDH samples during their thermal decomposition into the corresponding oxides. This suggests that the thermal decomposition of both LDH materials leads to mixed oxides whose surface is enriched in Al. On the other hand, the surface Ce/Al ratio decreases while the surface Ce^3+^/Ce^4+^ ratio increases during thermal decomposition of the co-precipitated LDH. Conversely, the surface Ce/Al ratio increases while the surface Ce^3+^/Ce^4+^ ratio decreases for the mechanochemically prepared oxide. This suggests the existence of some differences in the surface properties of Ce-PP and Ce-MC mixed oxides.

### 2.4. DRIFT Measurements

The DRIFT spectra of the solids obtained by two different preparation methods, co-precipitation (HTCe-PP) and mechanochemical (HTCe-MC), and their corresponding mixed oxides are represented in Figure 3. The dried and calcined samples prepared by mechanochemical mixing present the same absorption bands as those of the corresponding samples obtained by the co-precipitation method.

The DRIFT spectra of the reference samples HTCe-PP and HTCe-MC, Figure 3a, show at around 3556 cm^−1^ an intense band that corresponds to the OH group vibration, ν(O–H), in the hydrotalcite structure and a shoulder at approximately 3100 cm^−1^, suggesting the presence of a second type of –OH stretching vibration due to hydrogen bonding with the carbonate from the interlayer space. 

The band at 1624 cm^−1^ is characteristic of the H_2_O bending vibration of interlayer water in the hydrotalcite structure. The bands at 1415 cm^−1^ and 700 cm^−1^ were assigned to the CO_3_^2–^ group vibration, while those at 600−400 cm^−1^ correspond to M–O bonds. The presence of unidentate carbonate asymmetric vibration is highlighted at 1515 cm^−1^, while the peak at 944 cm^–1^ corresponds to carbohydrates. During calcination treatments, a part of the OH^–^ and CO_3_^2–^ groups are removed from the structure as carbon dioxide and water, leading to a decrease of the intensity of the bands at 3100 and at around 1600 cm^−1^, the latter suffering at the same time a shift from 1624 cm^−1^ (in the dried state) to 1616 cm^−1^ (in the mixed oxides state). Because the carbonate anion is more stable, it is not totally removed at temperatures below 700 °C and, hence, it is also present in the calcined samples. The structural change that appeared by calcination also generated shifting of the bands from 1515 cm^−1^ to 1484 cm^–1^, 1415 cm^–1^ to 1434 cm^–1^, and 944 cm^–1^ to 914 cm^–1^.

### 2.5. Raman Analysis

The Raman spectra of the dried samples (Figure 4) show, at 150–153 cm^−1^, one band attributed to the lattice vibrations [30]. The bands around 463 and 553 cm^−1^, attributed to the Mg–O–Mg and Al–O–Al symmetric stretching vibrations, are accentuated in the case of the dried samples. The band at 463 cm^−1^ overlaps with the band corresponding to pure CeO_2_ in the F_2g_ vibrational mode, displaying a symmetric breathing mode of four O^2−^ ions around each Ce^4+^ cation [31].

Also in the region of carbonate anions, there is a band at 1058 cm^−1^ which may be associated with the Mg_3_OH unit [30] that is shifted toward 1077 cm^−1^ in mixed oxide samples as the ratio of the divalent to trivalent cations decreases [32]. The band at 1341 cm^−1^ is for the vibrations of CO_3_^2–^ anions (v_3_ symmetric stretching) [33]. In the carbonate vibrational region, there are at least four visible bands, especially in the case of the calcined samples which could not lose all the carbonate since the calcination temperature was below 700 °C. These bands occur at 1355 and 1395 cm^−1^ (due to the CO_3_^2−^ antisymmetric stretching modes), at 1521 cm^−1^ (corresponding to the CO_3_^2–^ ions [34]), and at 1645 cm^−1^ (attributed to the hydrogen bonds created by the water molecules [30]). Only a band at 2872 cm^−1^ appears in the hydroxyl stretching region due to the strong hydrogen bonds obtained between the carbonate anion and water.

### 2.6. DRUV-VIS NIR Analysis 

The DRUV-VIS NIR spectra for the calcined compounds (Ce-PP and Ce-MC) are represented in Figure 5.

The diffuse reflectance spectra of solids containing CeO_2_ supported on alumina or silica present intense bands at 250 and 297 nm in the UV region, which are assigned to Ce^3+^ → O^2–^ and Ce^4+^ → O^2–^ charge transfer transitions, respectively [35,36]. In our case, for the calcined LDHs prepared by both methods, the Ce^4+^ → O^2–^ charge transfer transition peak is shifted with 17 nm to higher wavelengths (e.g., 314 nm), indicating a lower energy interaction. For both samples, there is also a peak at 290 nm, which is more intense for the sample Ce-MC, indicating a stronger charge transfer for this sample. Two absorption bands with similar intensities for the two samples were observed at higher wavelengths belonging to the NIR region, which can be ascribed to fundamental stretching frequencies of surface molecular groups such as H_2_O and O–H [37].

### 2.7. Determination of Acido-Bascity and Textural Properties

The surface areas, bulk and real densities, acidities, and basicities of the cerium-modified calcined hydrotalcite-like materials are tabulated in Table 4. It can be observed that the Ce-MC material has a total number of base sites two times higher than that of the Ce-PP sample. Moreover, while the Ce-MC sample show basic sites of all strengths, i.e., strong, medium, and weak, the Ce-PP sample exposes almost only strong basic sites. This implies that the Ce-MC catalyst presents higher densities of OH^–^ and O^2–^ anions on its surface. This fact can be well related to the higher number of defects induced in the structure when using the mechanochemical method which does not include ageing. On the other hand, both mixed oxides possess a certain amount of surface acidity, in line with the XPS data showing that their surface is enriched in Al. At the same time, the Ce-PP sample shows a significantly higher number of acid sites compared to Ce-MC. In the former, the Brønsted acid sites are dominant (76%), while the latter mostly shows Lewis acid sites (95.6%). Both real and bulk densities of the Ce-PP sample are about two times higher compared to those of the Ce-MC sample, suggesting a significantly better packing of the former. Both samples show large surface areas as already noticed for mixed oxides obtained by calcination of corresponding LDHs modified with rare earth cations [12].

### 2.8. Catalytic Activity Tests

#### 2.8.1. Self-Condensation

Table 5 presents the values of cyclohexanone conversion after 5 h under reflux conditions in the cyclohexanone self-condensation reaction, the products observed being 2-cyclohexylidencyclohexanone (A), 2-(1-cyclohexenyl)-cyclohexanone (A1), 2,6-dicyclohexylidencyclohexanone (B), and 2,6-di(1-cyclohexenyl)-cyclohexanone (B1). The catalytic activity of the mixed oxides is superior to that of the dried samples regardless of the preparation method used. This should be due to the fact that dried LDH materials possess only weak and moderate strength OH basic sites and not strong Lewis-base sites as their corresponding oxides [38]. Notably, the mechanochemical method generates slightly more active catalytic materials, likely due to their higher basicity and larger specific surface area as shown in Section 2.7 for the calcined mixed oxides. In all cases, the selectivity to the mono-condensation product A is high, within the domain of 76–88%, being larger for the calcined catalysts than for their corresponding LDHs. It is worth noting that the mixed oxide catalysts gave only mono-condensation products A and A1, the selectivity towards di-condensation products being zero. The latter remains quite small for the LDH catalysts as well.

In the presence of cyclohexane (5 mL) as a solvent (Entry 6), the activity of the Ce-MC catalyst is lowered without significant changes in the product distribution due to the competition between the reactant and solvent for the adsorption sites present on the catalyst surface. By decreasing the reaction temperature from reflux conditions towards 100 °C, the conversion values for the reaction catalyzed by Ce-MC lowered by 25%.

The stability of the Ce-MC catalyst was checked in four consecutive cyclohexanone self-condensation runs, the catalyst at the end of the reaction being separated by filtration and reused as such. After the four cycles, Entry 5, the conversion decreased by about 5%, thus confirming the stability of this material in the reaction. However, there was a slight change in the ratio between mono- and di-condensation products.

#### 2.8.2. Claisen-Schmidt Condensation

The conversions and selectivities obtained from one-pot synthesis of chalcone (CH) or flavone (FL) via Claisen–Schmidt condensation reactions between benzaldehyde (BA) and acetophenone (ACP) or benzaldehyde and 2′-hydroxyacetophenone (HAP) at different molar ratios (benzaldehyde/acetophenone or benzaldehyde/2′-hydroxyacetophenone = 1/1; 5/1; 10/1) are presented in Table 6. It can be observed that, as expected, by increasing the molar ratios of the reagents, higher values of conversion with lower selectivities are obtained. Moreover, the Ce-MC catalyst was proven to be more efficient than Ce-PP, likely due to the higher number of base sites and larger surface area (see Table 4) of the former. The higher activity of the sample prepared by the mechanochemical method can also be related to the fact that both HTCe-MC and Ce-MC samples had smaller crystallite sizes compared to their co-precipitated counterparts, as revealed by both the results of XRD analyses (Table 1 and Table 2) and SEM micrographs, since this preparation method did not imply the ageing step for the crystal growth. Notably, the conversions of 2′-hydroxyacetophenone (HAP) are higher than those reached with acetophenone (ACP) on both catalysts and the selectivities to flavone are lower than those for chalcone. This fact could be a consequence of the higher reactivity of 2′-hydroxyacetophenone compared to acetophenone.

## 3. Materials and Methods

### 3.1. Synthesis of The LDHs and Their Corresponding Mixed Oxides

The hydrotalcite-like compounds modified with cerium were prepared by the co-precipitation and mechanochemical method, respectively. To obtain the solid by the co-precipitation route, two solutions were used (A and B). Solution A was prepared by dissolving 0.0343 mol of Ce(NO_3_)_3_∙6H_2_O, 0.5209 mol of Mg(NO_3_)_2_∙6H_2_O, and 0.1395 mol of Al(NO_3_)_3_∙9H_2_O in 469 mL of distilled water while solution B resulted by dissolving 1.1575 mol of NaOH and 0.4632 mol of Na_2_CO_3_ in 387 mL of distilled water. The two solutions, A and B, were mixed in a round-bottomed flask under continuous stirring at 25 °C and kept for ageing at 70 °C for 18 h. The resulting gel was filtered and washed with distilled water until the conductivity of the wash water reached a value below 100 μS/cm. The resultant layered double hydroxide-type compound presents the following formula: Mg_3_Al_0.75_Ce_0.25_(OH)_8_(CO_3_)_0.5_∙2H_2_O. The solid obtained by co-precipitation was noted as HTCe-PP.

The same compound was also prepared by the mechanochemical route based on the dry grinding of the reagents in a mortar. Practically, the amounts of nitrates respecting the required molar ratios were mixed with sodium hydroxide and carbonate for 1 h, yielding a yellowish paste which was further washed as in the previous case. The prepared compound was named HTCe-MC. The two hydrotalcites prepared by both methods were further dried in an oven at a temperature of 90 °C for 24 h under an air atmosphere. The corresponding mixed oxides were obtained by calcination at 460 °C, for 18 h, and they were noted as Ce-PP (resulting from HTCe-PP) and Ce-MC (resulting from HTCe-MC).

### 3.2. Characterization Techniques

The XRD patterns of the obtained layered double hydroxides and their corresponding calcined forms were recorded using an X’Pert PANanalytical MPD system equipped with monochromatic CuKα radiation (λ = 1.5418 Å) with 0.02° (2θ) steps over a 5–70° 2θ angular range with a 10 s counting time per step. The structural data were evaluated using the HighScore software from Panalytical (Almelo, Netherlands).

Morphological characterizations of the samples were performed by scanning electron microscopy (SEM). SEM investigations were carried out using a scanning electron microscope (JSM-531 Inspect S Electron Scanning Microscope, FEI Company) at accelerating voltages of 10 kV. The chemical evaluation of the samples was performed using energy-dispersive X-ray spectroscopy (EDS) via an Element 2CB detector on the JSM-531 Inspect S Electron Scanning Microscope, FEI Company (Eindhoven, Netherlands). 

An Escalab Xi+ system (Thermo Scientific, Waltham, MA, USA) was used for the X-ray photoelectron spectroscopy (XPS) survey and high-resolution XPS spectra acquisition. The survey scans were acquired using an Al Kα gun with a spot size of 900 µm, pass energy of 10.0 eV, and an energy step size of 1.00 eV (five scans). Twenty scans were accumulated for the Ce3d high-resolution XPS spectra, the pass energy was set to 10.0 eV, and the energy step size was 0.10 eV. 

DRIFTS spectra were obtained after an accumulation of 400 scans for the 400–4000 cm^−1^ domain with a JASCO FT/IR-4700 LE spectrometer (Jasco, Tokyo, Japan). 

The Raman spectra were recorded in extended mode in the range of 100–3200 cm^−1^ with a spectral resolution of 1 cm^−1^, an acquisition time of 10 s, and 1 scan/point using a confocal Renishaw InVia Reflex Raman spectrometer (Renishaw, New Mills Wotton-under-Edge Gloucestershire United Kingdom) with a Co excitation source of 532 nm, having a diode pumped solid state (DPSS) laser at a power of 200 mW. The microscopy objective (20×) and the laser focalization were adjusted to visualize the microscopic image of the composites. 

The diffuse reflectance UV-VIS-NIR spectra were recorded using a UV3600 UV-VIS Shimadzu spectrometer (Shimadzu, Kyoto, Japan) and extra pure barium sulfate (BaSO_4_) as the white reference. The data were collected with 0.5 nm step and 8 nm slit width in the UV-Vis region and 2 nm step and 32 nm slit width in the NIR region.

The base sites distribution of the cerium modified compounds was determined by irreversible adsorption of organic acids with different values of pKa followed by quantitative determination of the adsorbed acid using a UV-VIS spectrometer (Jasco V650 Tokyo, Japan). Acrylic acid, with pKa = 4.2, was used to find the total number of basic sites, while phenol, pKa = 9.9, was utilized to determine the strong base sites. The number of low strength base sites was obtained from the difference between the total number of base sites and the number of strong base sites. Before basicity measurements, all catalysts were thoroughly calcined under static air conditions in an oven. The catalysts were weighed (0.05 g) and added in different volumetric flasks. Over the catalysts, 10 mL of freshly prepared solution of phenol/acrylic acid in cyclohexane was pipetted out and was shaken for 2 h keeping the flasks closed. It was assumed that the interaction of the catalysts with atmospheric CO_2_ and water was very negligible since the exposure of the samples to the atmosphere was for a concise period (during weighing only). The concentration of the substrate in solution in equilibrium with the adsorbed substrate was spectrophotometrically measured. Sorption experiments were developed at the wavelength of maximum adsorption corresponding to phenol (λ_max_ = 271 nm) and acrylic acid (λ_max_ = 225 nm), in the concentration range of the adsorbate where the Beer–Lambert law holds well. 

Pyridine adsorption was used for the determination of the total number of acid sites. The freshly calcined solids (0.05 grams) were wetted with pyridine aliquots (0.2 μL each) and maintained under inert flow at 90 °C for the removal of physiosorbed pyridine. The procedure was repeated until the weight of each solid was constant after two consecutive additions of pyridine. The amount of adsorbed pyridine was divided by its molar weight to obtain the number of mmol of adsorbed pyridine which is equal to the total number of acid sites determined by this method. To discern between the Brønsted and Lewis type acid sites, an FT/IR-4700 Jasco spectrometer (Tokyo, Japan) was used to record the DRIFT spectra of the solids with adsorbed pyridine considering the DRIFT spectra of the freshly dried materials as the background. According to literature data, the bands corresponding to pyridine adsorbed on Lewis acid sites appear in the ranges of 1435–1455 cm^−1^ and 1570–1615 cm^−1^, while those corresponding to pyridine adsorbed on Brønsted acid sites appear in the range of 1520–1555 cm^−1^ and at 1630 cm^–1^ [39,40,41]. The areas of the peaks appearing in each of these regions were integrated and their total was calculated. Afterwards, the percentage distribution of the peaks corresponding to each type of acid site was calculated.

Real and bulk densities of the prepared catalysts were also determined. The real density of the LDHs was experimentally established using the pycnometer method. The bulk density was determined by averaging the results of three consecutive weightings of known volumes of the solid samples. The available volumes of solid were measured using a graduated cylinder where the solid sample was poured in. The filled cylinder was mechanically tapped until no further volume change was observed, and then it was weighed. 

N_2_ adsorption-desorption isotherms were determined using a Micromeritics ASAP 2010 instrument (Micromeritics, Eindhoven, The Netherlands), where samples were outgassed under vacuum for 24 h at 120 °C before nitrogen adsorption.

### 3.3. Catalytic Tests

Each catalyst was finely ground in an agate mortar and the granulometric fraction (80–100 mesh) was used in each reaction. The catalytic activity of the prepared solids was tested first in the self-condensation reaction of cyclohexanone (Figure 1), which requires 10 mmol of organic compound and 5 wt.% catalyst under batch and solvent-free conditions, with magnetic stirring (200 rpm) at reflux (160 °C) for 5 h of reaction time. They were also tested in two Claisen–Schmidt condensation reactions (Figure 2 and Figure 3) with the aim of obtaining compounds belonging to the flavonoid family, namely chalcone (Figure 2) and flavone (Figure 3). For chalcone synthesis, benzaldehyde and acetophenone were used at different molar ratios, namely 1/1, 5/1, and 10/1; to obtain flavone, acetophenone was replaced with its derivative compound 2′-hydroxyacetophenone, keeping the same benzaldehyde-to-ketone molar ratios. All the reagents were purchased from Merck. In all reactions, the catalyst concentration was 5 wt.% in the reaction admixture (*w*/*w*). The reactions were performed in a 50 mL stirred flask (200 rpm), under reflux conditions (e.g., 205 °C for chalcone synthesis and 300 °C for flavone synthesis), for 4 h of reaction time.

All the condensation products were analyzed with a Thermo Quest Chromatograph from ThermoFischer Scientific, Massachusetts, US equipped with an FID detector and a capillary column of 30 m with a DB5 stationary phase. The reaction products were also identified by mass spectrometry-coupled chromatography, using a GC/MS/MS VARIAN SATURN 2100 T system (Varian Palo Alto, CA, USA) equipped with a CP-SIL 8 C.B. Low Bleed/M.S. column of 30 m length and 0.25 mm diameter.

## 4. Conclusions

The mechanochemical route of preparation has enabled the elimination of some of the steps, equipment, and energy consumption required in the co-precipitation method. The synthesized compounds exhibited the hydrotalcite structure, as indicated by the XRD measurements, whatever the preparation method used. However, apart from a ceria minor phase present in both, in the co-precipitated LDH material a supplementary side phase, i.e., cerium oxycarbonate, was identified. 

The mixed oxide derived from the LDH prepared by the mechanochemical method shows larger surface area, lower crystallite size, lower real and bulk densities, higher basicity, and lower acidity compared with that obtained from the co-precipitated LDH material, with consequences on its catalytic performance in the studied reactions. 

The LDH materials had comparable activities in cyclohexanone self-condensation, but their activities were significantly lower compared to those of their corresponding mixed oxides. Among the latter, the mechanochemically prepared oxide was slightly more active than its co-precipitated counterpart. In all cases, the mono-condensation product 2-cyclohexylidencyclohexanone was the main reaction product, with selectivities of 76–84% for the LDH catalysts, and 86–88% for the LDH-derived mixed oxides. The mechanochemically prepared oxide showed a good stability during four reaction cycles.

In the Claisen–Schmidt condensation reactions, the mechanochemically prepared oxide catalyst was also shown to be more efficient than its co-precipitated counterpart due to its higher number of base sites, larger surface area, and lower crystallite size. At the same time, both oxides show good selectivities to the desired product, with the mechanochemically prepared oxide catalyst being slightly more selective than its co-precipitated counterpart, probably due to its higher basicity and lower acidity. 

## Data Availability

The data are available on request from the corresponding author. Appendix A are included.

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
