# Peer review of "The Influence of the Preparation Method on the Physico-Chemical Properties and Catalytic Activities of Ce-Modified LDH Structures Used as Catalysts in Condensation Reactions"

_molecules, 2021, doi:10.3390/molecules26206191_

Round 1

Reviewer 1 Report

The impact of LDH's catalytic preparation method on catalytic activity is of great interest to many researchers.  I recommend that this manuscript can be published in "molecules" journal if the authors correct the following points.

     1.In abstract (p.1).  Which is correct, HTCe-Mc or HTCe-MC?

  1. On p.2. I can understand the advantages of mechanical treatment from the text, but not the disadvantages. Please specify the shortcomings and how to overcome them. It is a major premise that this overcoming method is used in this paper.

  1. The positions of Table 1 and Table 5 are incorrect. Isn't such a notation rude to the peer reviewers, even for peer review?

  1. What is “Ce-AM” or “CeAM”? Same as in Fig. 3 and Fig. 5!

  1. This manuscript is confusing because it has different notations for the same substance, such as HTCe-PP and HTCePP. Including this matter, please use the same notation in this manuscript in a unified manner.

  1. Please indicate the reason for the result shown in Fig. 2 by reflecting the characteristics of the preparation methods.

  1. Fig. 5 (a) and Table 4 show the largest difference in catalyst preparation methods. Please explain this result by reflecting the characteristics of the preparation methods.

  1. Please explain in detail which characteristics of the catalytic characterization obtained by both catalyst preparation methods are reflected in the results of the catalytic reactions (Table 5 and Table 6).

  1. Conclusion should be revised using discussion combining the results on characterization and catalytic reaction. In the present manuscript, there were no discussion but only results.

Author Response

The impact of LDH's catalytic preparation method on catalytic activity is of great interest to many researchers.  I recommend that this manuscript can be published in "molecules" journal if the authors correct the following points.

  We thank the reviewer for taking the time to evaluate our work. Our answers to the queries are presented below highlighted in yellow:

     1.In abstract (p.1).  Which is correct, HTCe-Mc or HTCe-MC?

we amended the text HTCe-MC

  1. On p.2. I can understand the advantages of mechanical treatment from the text, but not the disadvantages. Please specify the shortcomings and how to overcome them. It is a major premise that this overcoming method is used in this paper.

 MC method may have a shortcoming in what concerns the crystallinity of the resulting material. However from the point of view of the catalytic activity the presence of crystallinity defects may increase the number of active sites and their dispersion on the surface.

  1. The positions of Table 1 and Table 5 are incorrect. Isn't such a notation rude to the peer reviewers, even for peer review?

 We are sorry that the positions of Table 1 and Table 5 were not correct since the headline was split on 2 consecutive pages we have corrected that.

  1. What is “Ce-AM” or “CeAM”? Same as in Fig. 3 and Fig. 5!

We thank reviewer 1 for this observation. We have corrected the names on the figures (initially we have used AM as abbreviation for mechanochemical method (MC).

  1. This manuscript is confusing because it has different notations for the same substance, such as HTCe-PP and HTCePP. Including this matter, please use the same notation in this manuscript in a unified manner.

We thank reviewer 1 for this observation. We have checked and modified the abbreviations accordingly.

  1. Please indicate the reason for the result shown in Fig. 2 by reflecting the characteristics of the preparation methods.

We thank reviewer 1 for this suggestion We have amended the text as it follows:

The SEM images of HTCe-MC and Ce-MC showed agglomerations of smaller particles compared to the samples obtained by precipitation method. This fact may be related to the absence of the ageing step in the mechanochemical preparation which shortened the crystal growth duration and hence the size of the crystallites

  1. Fig. 5 (a) and Table 4 show the largest difference in catalyst preparation methods. Please explain this result by reflecting the characteristics of the preparation methods.

We thank reviewer 1 for this suggestion We have amended the text.

It can be observed that the Ce-MC material has a total number of base sites two times higher compared with the Ce-PP sample. Moreover, while the Ce-MC sample show basic sites of all strengths, i.e. strong, medium and weak, the Ce-PP sample exposes almost only strong basic sites. This implies that the Ce-MC catalyst presents higher densities of OH- and O2- anions on its surface. This fact can be well related to the higher number of defects induced in the structure when using the mechanochemical method which does not include ageing. On the other hand, both mixed oxides possess a certain amount of surface acidity, in line with the XPS data showing that their surface is enriched in Al. At the same time, the Ce-PP sample shows a significantly higher number of acid sites compared to Ce-MC. In the former the Brønsted acid sites are dominant (76 %), while the latter mostly shows Lewis acid sites (95.6 %).

 The discussion of the XPS data in Table 3 was also modified as it follows:

It is noteworthy that the surface Mg/Al ratio decreases for both LDH samples during their thermal decomposition into the corresponding oxides. This suggests that the thermal decomposition of both LDH materials leads to mixed oxides whose surface is enriched in Al. On the other hand, the surface Ce/Al ratio decreases while the surface Ce3+/Ce4+ ratio increases during thermal decomposition of the coprecipitated LDH. Conversely, the surface Ce/Al ratio increases while the surface Ce3+/Ce4+ ratio decreases for the mechanochemically prepared oxide. This suggests the existence of some differences in the surface properties of Ce-PP and Ce-MC mixed oxides.

  1. Please explain in detail which characteristics of the catalytic characterization obtained by both catalyst preparation methods are reflected in the results of the catalytic reactions (Table 5 and Table 6).

Answer:

For the results in Table 5 we had already included in the text: “Notably, the mechanochemical method generates slightly more active catalytic materials, likely due to their higher basicity as shown in Section 2.7 for the calcined mixed oxides.” We have also ammended this comment including “and larger surface area”.

For the results in Table 6 we had already included in the text Its higher catalytic activity can be well correlated to its higher number of base sites (see Table 4).

Now we revised the paragraph as follows: It can be observed that, as expected, by increasing the molar ratios of the reagents, higher values of conversion but with lower selectivities are obtained. Moreover, the Ce-MC catalyst has proven to be more efficient than Ce-PP, likely due to the higher number of base sites and larger surface area (see Table 4) of the former. The higher activity of the sample prepared by the mechanochemical method can also be related to the fact that both HTCe-MC and Ce-MC samples had smaller crystallite sizes compared to their coprecipitated counterparts, as revealed by both the results of XRD analyses (Tables 1 & 2) and SEM micrographs since this preparation method did not imply the ageing step for the crystal growth.

  1. Conclusion should be revised using discussion combining the results on characterization and catalytic reaction. In the present manuscript, there were no discussion but only results.

The results were further commented and the Conclusion has been revised accordingly.

Once again we would like to thank the reviewer for carefully reading our manuscript and suggesting significant changes to improve its quality.

On behalf of all authors:

Associate Professor dr. Rodica Zavoianu

Professor dr. Ioan-Cezar Marcu

Reviewer 2 Report

1 – Authors must not use decimal places when mentioning BET surface areas. For instance, in Table 4, 199,5 m2/g;

2 – Details on the particle size of the catalysts must be provided. It is important to indicate that diffusion limitations inside the catalyst particles were eliminated.

3 – Agitation speed was not provided. It is important to indicate that external mass transfer resistance was eliminated owing to the stirring speed.

Author Response

1 – Authors must not use decimal places when mentioning BET surface areas. For instance, in Table 4, 199,5 m2/g;

We have corrected 200 m2/g and 184 m2/g

2 – Details on the particle size of the catalysts must be provided. It is important to indicate that diffusion limitations inside the catalyst particles were eliminated.

We specified in Section 3.3 that each catalyst was finely ground in an agate mortar before the reaction and the granulometric fraction 80-100 mesh was used.

3 – Agitation speed was not provided. It is important to indicate that external mass transfer resistance was eliminated owing to the stirring speed.

The stirring speed was also specified in Section 3.3.

Briefly the text in section 3.3. was modified

Each catalyst was finely ground in an agate mortar and the granulometric fraction (80-100 mesh) was used in each reaction. The catalytic activity of the prepared solids was tested first in the self-condensation reaction of cyclohexanone (Scheme 1), …………….., with magnetic stirring (200 rpm) … ……. They were also tested in two Claisen-Schmidt condensation reactions (Scheme 2), …………………………………… in 50 mL stirred flask (200 rpm), …………………

We would also like to thank the reviewer for appreciating our manuscript and suggesting significant changes to improve its quality.

On behalf of all authors:

Associate Professor dr. Rodica Zavoianu

Professor dr. Ioan-Cezar Marcu

Round 2

Reviewer 1 Report

I have confirmed that the authors are sincere in responding to my questions and suggestions. I recommend this revised manuscript to be published in Molecules.